# Speech Separation based on pre-trained model and Deep Modularization

## Abstract

Deep neural networks (DNN) have been used extensively to achieve impressive results in speech separation. Most of the DNNs implementations to speech separation relies on supervised learning which is data hungry, and success is pegged on availability of large-scale parallel clean-mixed speech pair. This kind of data is always not available since it is difficult to create hence limiting the implementation of supervised learning. Moreover, the implementation of supervised learning in speech separation requires that systems deal with the permutation problem (permutation ambiguity). This places an upper limit of the quality of separated speech that a tool can attain. To avoid the problem of permutation ambiguity, speech separation based on clustering has been proposed by some recent works. However, these clustering techniques still rely on supervised learning and therefore still require quality paired data. To deal with the problem permutation ambiguity and eliminate need for paired training dataset, we propose a fully unsupervised speech separation technique based on clustering of spectrogram points or raw speech blocks. Our technique couples the traditional graph clustering objectives and deep neural networks to achieve speech separation. We start by establishing features of spectrogram points or raw speech blocks using a pre-trained model and consequently use the features in a downstream task of clustering using deep modularization. Through this we are able to identify clusters of spectrogram points or raw speech blocks dominated all speakers in a mixed speech. We perform extensive evaluation of the proposed technique and show that it outperforms state of the art tools included in the study.

## 1 Introduction

Speech separation involves isolating each independent speech composed in a mixture speech. For a mixture $y(n)$ that is composed of $C$ independent speech signals $x_c(n)$ with $c = 1, \cdots, C$, $y(n)$ can be represented as:

$$y(n) = \sum_{c=1}^{C} x_c(n) \tag{1}$$

Where $n$ indexes time. Separating speech from another speech is a daunting task by the virtue that all speakers belong to the same class and share similar characteristics Hershey et al. (2016a). Some tools such as Wang et al. (2017) and Wang et al. (2016) develop techniques that perform speech separation on a mixed speech signal based on gender voices present. They leverage the large discrepancy between male and female voices in terms of vocal track, fundamental frequency contour, timing, rhythm, dynamic range to attain gender-based speech separation. In speech separation tasks where the mixture is composed of speakers of the same gender, the separation task is much difficult since the pitch of the voice is in the same range Hershey et al. (2016a). To perform speech separation in such cases, most tools such as Zeghidour & Grangier (2021a) Huang et al. (2011) Weng et al. (2015) Isik et al. (2016) Hershey et al. (2016a) and Luo & Mesgarani (2019a) cast the task as a multi-class regression problem. In that case, training a speech separation model involves comparing its output to a source speaker and the model will always output a dimension for each target class. When multiple sources of the same type exist, the system needs to select arbitrarily which output

dimension to map to each target and this raises a permutation problem (permutation ambiguity) Hershey et al. (2016a). Therefore, systems that perform speaker separation in this manner have an extra burden of designing mechanisms that are geared towards handling the permutation problem. The standout technique for tackling permutation ambiguity problem is the permutation invariant training (PIT) technique Yu et al. (2017) and Kolbæk et al. (2017). The key strategy in PIT is to determine the best output-target pairing and perform error minimization based on this pairing. While effective, the technique has been criticised for having a high computation complexity of $O(S!)$ which is computationally expensive when the number of sources $S$ is high Tachibana (2021) Dovrat et al. (2021). PIT based speech separation models must also deal with the problem of mismatch between number of speakers in training and inference due to their fixed output dimension Jiang & Duan (2020). Models have to design creative techniques such as setting a maximum number of sources $S$ that the model should output from any given mixture Nachmani et al. (2020) Kolbæk et al. (2017) Liu & Wang (2019a) Luo & Mesgarani (2020). If an inference mixture has $K$ sources, where $S > K$, $S - K$ outputs are invalid sources. In case of invalid sources, some models such as Liu & Wang (2019a), Nachmani et al. (2020), Kolbæk et al. (2017) output silences while some such as Luo & Mesgarani (2020) output the mixture itself which are then discarded by comparing the energy levels of the outputs relative to the mixture. The models that output silences for invalid sources, rely on a pre-defined energy threshold, which may be problematic if the mixture also has a very low energy Luo & Mesgarani (2020). Some models such Shi et al. (2018), Kinoshita et al. (2018), Takahashi et al. (2019)Neumann et al. (2019) handle the output dimension mismatch problem by generating a single speech in each iteration and subtracting it from the mixture until no speech is left. However, setting the iteration termination criteria is difficult and the separation performance degrade in later iterations Takahashi et al. (2019). To evade the permutation ambiguity problem, works in Hershey et al. (2016b) Byun & Shin (2021) Isik et al. (2016) Qin et al. (2020) and Lee et al. (2022) Zeghidour & Grangier (2021b) propose a clustering technique that seeks to identify the multiple speakers present in a mixed speech signal. In these models, the deep neural network (DNN) $f_\theta$(parameterized by $\theta$) takes as its input a whole mixed speech spectrogram $X$ and generates a $d$ dimension embedding vector $V$ of size $N$ i.e., $V = f_\theta(X) \in R^{N \times d}$. The embedding $V$ learns the features of the spectrogram $X$ and is considered a permutation-and-cardinality-independent encoding of the network's estimate of the signal partition. Each $v_i \in V$ is such that $|v_i|^2 = 1$ for $i = 1, \cdots, N$. The embedding $V$ is considered to implicitly represent an $N \times N$ estimated affinity matrix $\hat{A} = VV^T$. For the network $f_\theta$ to learn how to generate an embedding vector $V$ given the input $X$, it is trained to minimize the cost function.

$$C_Y(V) = ||\hat{A} - A||^2 = ||VV^T - YY^T||_F^2 = \sum_{ij}(< v_i, v_j > - < y_i, y_j >)^2 \qquad (2)$$

The goal in equation 2 is to minimise the distance between the network estimated affinity matrix $VV^T$ and the true affinity matrix $YY^T$. The minimization is done over the training examples. $||.||_F^2$ is the squared Frobenius norm. Once $V$ has been established, its rows $v_i$ of $V$ are clustered using $k$-means clustering algorithm. The resulting clusters of $V$ are then used to construct a binary mask which is applied to the mixed spectrogram $X$ to separate the sources. Despite the impressive results of the clustering based techniques such as in Zeghidour & Grangier (2021b), they still employ supervised training which require a costly process of data labelling. Furthermore, these methods require that the number of speakers to be known before execution which may not be practical in some cases. Our work proposes a speech separation technique based on clustering known as deep modularization that requires no parallel dataset and prior knowledge of number of sources present in the mixture. The clustering is done on the spectrogram points or raw speech blocks. Deep modularization bridges the gap between traditional graph clustering objectives and deep neural networks. We start by establishing features of spectrogram points or raw speech blocks using a pre-trained model and use the features in downstream task of clustering using deep modularization. Through this we achieve speech separation in a purely unsupervised manner. Our contributions are as follows: (i) We propose an unsupervised clustering technique of speech separation based on optimization of cluster assignments of spectrogram points or raw speech blocks in an end-to-end differentiable manner, (ii) we perform evaluation comparing the performance of the proposed technique when raw speech and Fourier transformed speech is used as input, (iii) we perform extensive evaluation of the proposed technique on different datasets to evaluate its suitability in speech separation in different scenarios, (iv) we perform ablation comparing the proposed technique of deep modularization and the classical $k$-means clustering technique and (v) we perform ablation

to evaluate if training the pre-trained model with contaminated inputs is beneficial to the downstream task of speech separation.

## 2    Related work

A number of unsupervised speech separation tools have previously been developed. MixIT technique Wisdom et al. (2020) performs unsupervised speech separation such that given a set of $X$ that is composed of mixed speeches i.e. $X = \{x_1, x_2, \cdots, x_n\}$ where each mixture $x_i$ is composed of up to $N$ sources, mixtures are drawn at random from the set $X$ without replacement and a mixture of mixture (MoM) created by adding the drawn mixtures, for example if two mixtures $x_1$ and $x_2$ are drawn from the set $X$, $MoM$ $\bar{x}$ is created by adding $x_1$ and $x_2$ i.e $\bar{x} = x_1 + x_2$. The MoM $\bar{x}$ is the input to a DNN model which is trained to estimate sources $\hat{s}$ composed in $x_1$ and $x_2$. The DNN model is trained to minimize the following loss function:

$$L_{MixIT} = \min_{A} \sum_{i=1}^{2} L(x_i, [A\hat{s}]_i)$$

For a case where MoM is composed of only two mixtures, $A \in B^{2 \times M}$ is a set of binary matrices where each column sums to 1. Here, $M$ is the number of maximum sources co-existing in a given mixture $x_i$. The loss function is trained to minimize the loss between mixtures $x_i$ and the remixed separated sources $A\hat{s}$. One major problem with the MixIT technique is that it performs over-separation such that the estimated sources are greater than the actual number of underlying sources in the mixtures $x_i$ Zhang et al. (2021). Moreover, MixIT does not work well for denoising Saito et al. (2021). A number of MixIT based techniques such as Zhang et al. (2021) and Karamatlı & Kırbız (2022)have been proposed to tackle the over-separation problem. To enable MixIT technique to handle denoising, work in Trinh & Braun (2022) and Saito et al. (2021) have been proposed. RemixIT Tzinis et al. (2022) is another unsupervised speech denoising tool that exploits teacher-student DNN model to perform speech separation. Given a batch of noisy speeches of size $b$, the teacher estimates the clean speech sources $\hat{s}_i$ and noises $\hat{n}_i$ where $1 \leq i \leq b$. The teacher estimated noises $\hat{n}_i$ are mixed at random to generate $n^p$. The mixed noise $n^p$ together with the teacher estimated sources are used to generate new mixtures $\hat{m}_i = \hat{s}_i + n^p$. The new mixtures $\hat{m}_i$ are used as input to the student. The student is optimised to generate $\hat{s}_i$ and noise $n^p$ i.e., $\hat{s}_i$ and $n^p$ are the targets. Through this, the teacher-student model is trained to denoise the speech. In RemixIT, a pre-trained speech enhancement model is used as the teacher model. Work in Wang et al. (2016) proposes unsupervised techniques to perform speech separation based on gender. They exploit i-vectors to model large discrepancy in vocal tract, fundamental frequency contour, timing, rhythm, dynamic range, etc between speakers of different genders. In this case DNN model can be viewed as gender separator.

To train these speech separation tools, models are developed to either accept Fourier spectrum or time domain input features. Fourier spectrum-based features tools do not work directly on the raw signal (i.e., signal in the time domain) rather they incorporate the discrete Fourier transform (DFT) in their signal processing pipeline mostly as the first step to transform a time domain signal into frequency domain. These models recognise that speech signals are highly non-stationary, and their features vary in both time and frequency. These features include Log-power spectrum features (Fu et al., 2017) (Xu et al., 2015) , Mel-frequency spectrum features (Liu et al., 2022) (Du et al., 2020) (Donahue et al., 2018), DFT magnitude features (Nossier et al., 2020) Grais & Plumbley (2018) Fu et al. (2019) and Complex DFT features (Fu et al., 2017) (Williamson & Wang, 2017) (Kothapally & Hansen, 2022a) (Kothapally & Hansen, 2022b). The assumption made by most DNN models that use Fourier spectrum features is that phase information is not crucial for human auditory. Therefore, they exploit only the magnitude or power of the input speech to train the DNN models to learn the magnitude spectrum of the clean signal and the factor in the phase during the reconstruction of the signal (Xu et al., 2014) (Kumar & Florencio, 2016) (Du & Huo, 2008) (Tu et al., 2014) (Li et al., 2017). The use of the phase from the noisy signal to estimate the clean signal is based on works such as (Ephraim & Malah, 1984) that demonstrated that the optimal estimator of the clean signal is the phase of the noisy signal. However, recent research Paliwal et al. (2011) have demonstrated through experiments that further improvements in quality of estimated clean speech can be attained by processing both the short-time phase and magnitude spectra. Due to phase challenge while working with Fourier spectrum features

different tool such as (Luo & Mesgarani, 2018) (Luo et al., 2020) (Luo & Mesgarani, 2019a) (Subakan et al., 2021a) explore the idea of designing a deep learning model for speech separation that accepts speech signal in the time-domain. The fundamental concept of these models is to replace the DFT-based input with a data-driven representation that is jointly learnt during model training. The models therefore accept as their input the mixed raw waveform sound and then generate either the estimated clean sources or masks that are applied on the noisy waveform to generate clean sources. By working on the raw waveform, these models address the key limitation of DFT-based models, since the models are designed to fully learn the magnitude and phase information of the input signal during training Luo et al. (2020).

## 3 Deep modularization speech separation model

### 3.1 Model pretraining

#### 3.1.1 Using spectrogram points as input

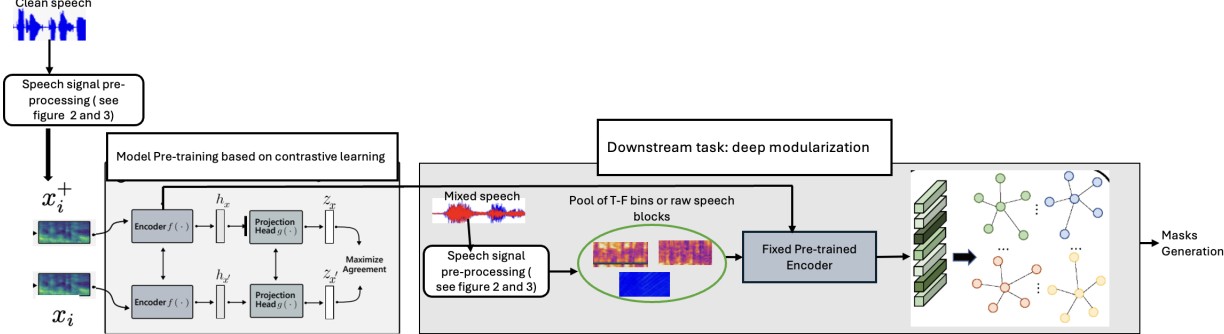

Figure 1: Overall overview of the proposed speech separation based on deep modularization

Our model first establishes a pre-trained model $f$ that learns to generate embeddings of spectrogram points (Time-Frequency bins (T-F bins)). We implement the contrastive learning similar to the one proposed in Saeed et al. (2021) but using spectrogram points (T-F bins) as the inputs. The goal of contrastive self-supervised learning is to establish a representation function $f : x \mapsto R^d$ that maps augmentations to a $d-$dimensional vectors by ensuring that similar view of augmentations are closer to each other as compared to those of random ones. The practice is to pick augmentations $(x, x^+)$ that are obtained by passing a given input through two different augmentation functions. Ideal augmentations of inputs are those that retain features of the inputs that are crucial in the intended task (e.g., classification) but modify the features that are less important for that task. Unlike work in Saeed et al. (2021) which does not use explicit augmentations, our work generates positive pairs by injecting both noise and reverberation to the input. Hence, given a clean speech signal from a given speaker in the time domain $x \in R^T$, we create its first augmented version in the time domain by adding randomly sampled excerpt from noise recorded in various urban setting from (Wichern et al., 2019). Its second augmented version is created by adding reverberation to the first augmented version using edited scripts from (Maciejewski et al., 2020)( see Figure 2). Each of the two versions is then downsampled to 8kHz and T-F bins generated from the magnitude spectrum by applying short-time Fourier transform (STFT) using 32 ms Hamming window and 8 ms shift. Two T-F bins that are extracted from both augmented sets belonging to a given speaker are considered a positive pair. It is important to note that since we are working at the T-F bins level, it is impractical to generate augmentations by adding add noise or reverberation to the T-F bins directly hence the indirect approach. Given a set $O$ of augmented T-F bins, we then design a function $f : O \mapsto R^d$ that maps the T-F bins to $d$-dimensional vectors by encouraging the representations of pairs of T-F bins from a positive pair to be closer to each other than the representations of T-F bins from another random speaker as demonstrated in figure 1. Therefore, given $n$ clean speeches from $n$ different speakers, we generate two sets $\bar{X}_A$ and $\bar{X}_B$ containing noise and noise+reverberation T-F bin augmentations as described above. We then exploit a function $\mathcal{P}(., . \mid \bar{X}_A, \bar{X}_B)$

that generates an augmentation pair $(x_i, x_i^+)$ by selecting T-F bins from both $\bar{X}_A$ and $\bar{X}_B$ i.e.

$$(x_i, x_i^+) \sim \mathcal{D}_{pos} \equiv (x_i, x_i^+) \sim \text{i.i.d } \mathcal{P}(.,. \mid \bar{X}_A, \bar{X}_B) \tag{3}$$

Here the pair $(x_i, x_i^+)$ is the positive pair with distribution $\mathcal{D}_{pos}$. Given a batch of size $b$, for a positive pair $(x_i, x_i^+)$, we consider all the other $b-2$ to be the negative examples with a distribution of $\mathcal{D}_{neg}$. To train the model to fit the function $f$, we use adopt simCLR contrastive loss Chen et al. (2020).

$$\mathbb{E}_{x,x^+\sim D_{pos}, x_{i:n-2}^-\sim D_{neg}} \left[ -\log\left( \frac{e^{f(x)^T f(x^+)}}{e^{f(x)^T f(x^+)} + \sum_{i=1}^{n-2} e^{f(x)^T f(x_i^-)}} \right) \right] \tag{4}$$

The loss function seeks to make the similarity $f(x)f(x^+)$ larger as compared to $f(x)f(x^-)$. Once the model is trained to establish T-F bin level features, we exploit the trained model in the downstream task.

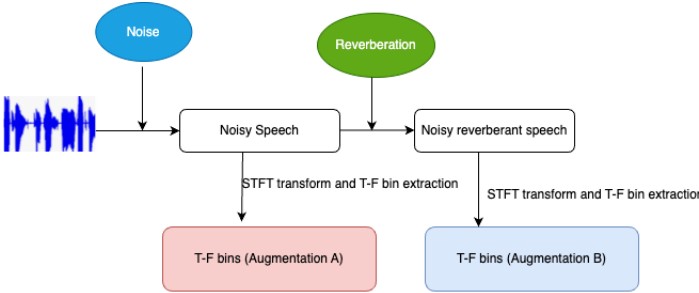

Figure 2: Creating T-F bins augmentations from a speech by first adding noise then reverberation.

### 3.1.2   Using time domain blocks as input

Given clean speech signal from a given speaker in the time domain $x \in R^T$, similar to discussion in section 3.1.1, we create its first augmented version in the time domain by adding randomly sampled excerpt from noise. The second augmented version is created by adding reverberation to the first augmented version. Each of the augmented speech version in the time-domain $x \in R^T$ is then processed by an encoder similar to the one proposed in Subakan et al. (2021b) to generate STFT-like representation $h \in R^{F \times T}$. This is then chunked into $N$ blocks of size 250 with 125 overlap between two subsequent blocks along the time axis to generate a set of blocks $L \in R^{F \times W \times N}$ (see figure 3). Therefore, given $n$ clean speeches from $n$ different

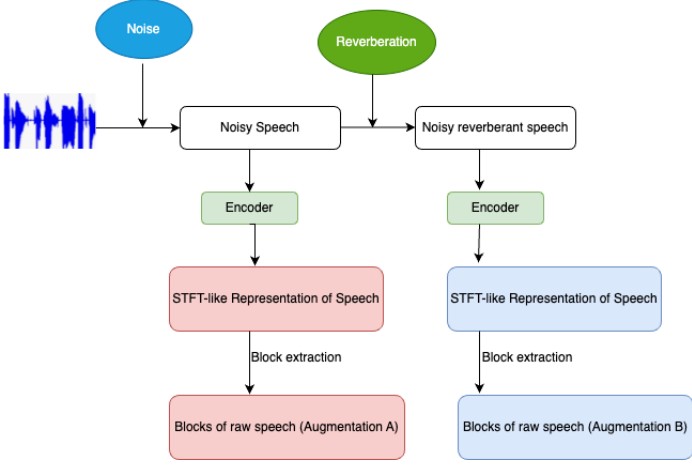

Figure 3: Creating T-F bins augmentations from a speech by first adding noise then reverberation.

speakers, we generate two sets $\bar{X}_A$ and $\bar{X}_B$ containing noise and noise+reverberation time domain speech blocks. We then exploit a function $\mathcal{P}(.,. \mid \bar{X}_A, \bar{X}_B)$ that generates an augmentation pair $(x_i, x_i^+)$ by selecting blocks from both $\bar{X}_A$ and $\bar{X}_B$ belonging to the same speaker. For a batch size of $b$, a pre-trained model based on raw speech blocks is trained similar to that described in section 3.1.1 and demonstrated in figure 1.

### 3.1.3 Pre-trained model

For the pre-trained models, we used EfficientNet-B0 (Tan & Le, 2019) as the encoder. EfficientNet-B0 is a lightweight-highly scalable convolutional neural network which was designed to accept 2D image inputs. Since the inputs i.e. T-F bins and raw speech blocks are in 2D, we did not make any modification to the EfficientNet-B0 and used it as originally proposed. In the last layer, we implemented a global max pooling to get an output embedding of $h \in R^{1280}$. When pre-training, we further processed $h$ using a projection head which is a fully connected feed forward layer with 512 units followed by a Layer Normalization and a tanh activation. We discarded the projection head during the downstream task.

### 3.2 Downstream task

The goal here is to use deep modularization technique to cluster T-F bins or raw speech blocks such that those dominated by a given speaker are clustered together. Deep modularization is a technique that seeks to intersect graph clustering objective and DNN. We begin by defining a graph $G(V, E)$ where $V = (v_1, v_2, \cdots, v_n)$, $|V| = n$ is the set of all T-F bins or raw speech blocks generated from a mixed speech signal as described in section 3.1.1 and 3.1.2. $E \subset V \times V$, $|E| = m$ is the set of all edges of connecting the generated T-F bins or raw speech blocks (subsequently referred to as blocks). We denote the adjacency matrix of $G$ by $A$ where $A_{ij} = 1$ if $\{v_i, v_j\} \in E$ and 0 otherwise. The degree of $v_i$ is defined as $d_i = \sum_j^n A_{ij}$, we are interested in generating a graph partition function $\mathcal{F} : V \to \{1, \cdots, k\}$ that splits the set of T-F bins or blocks $V$ into $k$ partitions $v_i = \{v_j : \mathcal{F}(v_j) = i\}$ given the T-F bins or blocks attributes $\bar{F} \in R^{n \times d}$ generated by the pre-trained model. In order to partition the vertices, we explore the statistical approach of vertices partitioning known as modularity (Q) (Newman, 2006). Modularity involves comparing the number of edges within partitions and some equivalent randomized partitions (null network) in which edges are placed without regard to relationships that exist in the network. modularity is, therefore, defined as:

$$Q = \text{Number of edges within partitions} - \text{expected number of such edges} \qquad (5)$$

A high value of $Q$, indicates closer similarities among members belonging to a given partition. Therefore, the goal is to maximise $Q$. Defining $g_i$ to be the community to which a vertex $i$ belongs to, Modularity (Q) is derived in Newman (2006) as:

$$Q = \frac{1}{2m} \sum_{ij} (A_{ij} - P_{ij}) \delta(g_i, g_j) \qquad (6)$$

where $\delta(g_i, g_j)$ is 1 if vertex $i$ and $j$ belong to the same partition and 0 otherwise. $P_{ij}$ is the expected number of edges between $i$ and $j$ while $A_{ij}$ is the actual number of edges between $i$ and $j$. If vertex $i$ and $j$ have degrees $d_i$ and $d_j$, respectively, then the expected degree of vertex $i$ can be defined as $\sum_j P_{ij} = d_i$. Based on this, vertex $i$ and $j$ are connected with probability $P_{ij} = \frac{d_i d_j}{2m}$ (see (Newman, 2006)). Hence equation 6 is modified to:

$$Q = \frac{1}{2m} \sum_{ij} (A_{ij} - \frac{d_i d_j}{2m}) \delta(g_i, g_j) \qquad (7)$$

The problem of maximizing $Q$ is NP-Hard (Brandes et al., 2006), however, if we seek to generate $k$ non-overlapping partitions, a partition assignment matrix $S \in R^{n \times k}$ ($n$ represents number of vertices) is defined (Newman, 2006). Each column of $S$ indexes a partition, that is, $S = \{s_1 \mid s_2 \mid, \cdots, \mid s_k \mid\}$. The columns are vectors of (0,1) elements such that $S_{ij} = 1$ if vertex $i$ belongs to partition $j$ and 0 otherwise. Based on this setup the columns of $S$ are mutually orthogonal since each row of the matrix sums to 1. $S$ therefore satisfies the normalization $Tr(S^T S) = n$ where $Tr(.)$ is the matrix trace. Based on the definition of $S$,

$\delta(g_i, g_j) = \sum_{k=1}^{k} S_{ik} S_{jk}$ and hence

$$Q = \frac{1}{2m} \sum_{ij=1} \sum_{n=1}^{k} (A_{ij} - P_{ij}) S_{ik} S_{jk} = \frac{1}{2m} Tr(S^T B S) \tag{8}$$

where $B$ is the modularity matrix such that $B_{ij} = A_{ij} - P_{ij}$. By relaxing $S \in R^{n \times k}$, the optimal $S$ that maximizes $Q$ is the top $k$ eigenvectors of matrix $B$. In our case, we seek to optimize $Q$ ( learn and optimize cluster assignment matrix $S$), by modularizing the T-F bin or block features $\bar{F} \in R^{n \times d}$ learned via the pre-trained model. To optimize the cluster assignment, we adapt the deep neural network graph partition technique proposed in Bianchi et al. (2020) and Müller (2023). They partition nodes of a graph by the following formulation:

$$\bar{F} = \text{GNN}(\tilde{A}, X, \theta_{GNN}) \tag{9}$$

$$S = \text{softmax}(\bar{F}) \tag{10}$$

Where $\tilde{A} = D^{-\frac{1}{2}} A D^{-\frac{1}{2}}$, $X$ are the input features, $D$ is the diagonal matrix with the degrees $d_1, \cdots, d_n$ on the diagonal and $A$ is the adjacency matrix. In equation 9, node features $\bar{F}$ are learned via graph neural network (GNN) and the assignment matrix S is established via SoftMax activation function. In (Bianchi et al., 2020), the assignment matrix $S$ is established by multilayer perception (MLP) with SoftMax on the output layer. In our case, we formulate the problem as:

$$\bar{F} = \text{Con}(X, \theta_{con}) \tag{11}$$

$$S = \text{SepFormer}(\bar{F}, \theta_{trans}) \tag{12}$$

Where the T-F bin or block feature matrix $\bar{F}$ is established via a pre-trained model (Con). The partition assignment of a T-F bin or block is established using a SepFormer proposed in (Subakan et al., 2021b). SepFormer is a transformer-based model that is able to learn both short-term and long-term dependencies that exist within a T-F bin or a block. The output of the SepFormer is passed through a feedforward network equipped with a softmax that assigns a T-F bin or a block to a given partition ( see figure 4). This essentially

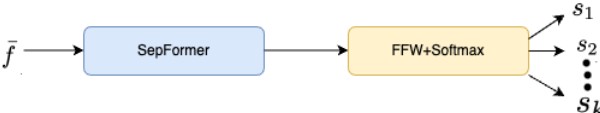

Figure 4: How SepFormer is utilized to perform T-F bin or block partitioning.

maps each T-F bin or block's feature $\bar{f}_i \in \bar{F}$ (with $1 \leq i \leq n$) to the $j$ column of the cluster assignment matrix $S$. To optimise the assignment $S$, we use the loss function in Equation 13 (Müller, 2023). By training the SepFormer using the objective in equation 13, we train a DNN model using a graph clustering objective. The loss in equation 13 is composed of a modularity (derived in Equation 8) term and a collapse regularizer. The collapse regularizer is crucial to avoid $S$ generating trivial partitions (Müller, 2023). Furthermore, it has been shown in Müller (2023) that the loss function in equation 13 maintains consistency of community detection as the number of nodes increases.

$$L(S) = -\frac{1}{2m} Tr(S^T B S) + \frac{\sqrt{k}}{n} || \sum_i S_i^T ||_F - 1 \tag{13}$$

Here, $||.||_F$ is the Frobenius norm. The complexity of the modularity term $Tr(S^T B S)$ is $\mathcal{O}(n^2)$ per update of $L(S)$ which makes the training process computationally costly. Therefore, to efficiently update $L(S)$, Müller (2023) proposes to decompose $Tr(S^T B S)$ into sum of sparse matrix-matrix multiplication and rank one degree normalization $Tr(S^T A S - S d^T d S)$. This reduces the complexity to $(O)(d^2 n)$ for every update of the loss function.

$$L(S) = -\frac{1}{2m} Tr(S^T A S - S d^T d S) + \frac{\sqrt{k}}{n} || \sum_i S_i^T ||_F - 1 \tag{14}$$

During implementation our model uses the loss in equation 14.

### 3.3 Adjacency matrix

To construct the adjacency matrix $A$, for each T-F bin or block $i$ we compute its similarity with all other nodes using inner product i.e.

$$e_{ij} = \bar{f}_i^T \bar{f}_j \tag{15}$$

where $j = 1, 2, \cdots, n$ and $\bar{f}_i$ and $\bar{f}_j \in \bar{F}$. We then select a threshold $\theta$ such that if $e_{ij} < \theta$, we remove an edge between $i$ and $j$ then the adjacency matrix is defined as

$$A_{ij} = \begin{cases} 1, & \text{if there is an edge between } i \text{ and j} \\ 0, & \text{otherwise} \end{cases} \tag{16}$$

Optimum $\theta$ is established experimentally (explained in section 6.6).

## 4 Clean source signal estimation

From the established partitions $k$, we generate $k$ masks in the range $[0, 1]$, where 0 indicates that a given T-F bin or block in the input mixed signal is missing in that cluster, while 1 signifies the presence of a given T-F bin or block. The mask-based separation of sources is predicated on the assumption of sparsity and orthogonality of the sources in the mixed signal in the domain in which masks are computed. Based on this assumption, the dominant signal at a given range is taken to be only signal at that range. Therefore, the generation of clusters through modularization is used to estimate the dominant signals in a given range. Once the masks have been established, they are applied to the input mixed signal to generate $k$ estimated clean signals. For the input speech signal that has been transformed to STFT, the mask is applied to the input STFT spectrogram to obtain the estimated spectrograms of clean speech signals. The inverse STFT is then used to estimate a clean speech signal. In case of a time domain signal, the mask is applied to the STFT-like transformation generated by the encoder. The decoder (transposed encoder) is the used to generate estimated signal. For STFT phase reconstruction, we use the technique proposed in Wang et al. (2018) which jointly reconstructs the phase of all sources in each mixture by exploiting their estimated magnitudes and the noisy phase using the multiple input spectrogram inversion (MISI) algorithm (Gunawan & Sen, 2010).

## 5 Experimental setup

**Pre-training dataset:** To pre-train the two model variants (i.e. one trained using T-F bins as input and another with raw speech blocks), we used the popular Wall Street Journal (WSJ0) corpus (Paul & Baker, 1992). The dataset was recorded using a close-talk microphone hence free from reverberation and noise. We used 30 hours of speeches from $si\_tr\_s$ to train the models. 30 hours length of speeches generate over 10 million T-F bins or blocks input dataset.
**Pre-training configuration**: To train the two variants of pre-trained models, we used the Adam optimiser and the cyclical learning rate (Smith, 2017) with a minimum learning rate of $1e-4$ and a maximum of $1e-1$. Each model was trained with a single NVDIA V100 GPU for 2M steps with a batch size of 512 T-F bins or blocks.
**Speech separation dataset**: To evaluate the quality of separated speech resulting from the proposed technique, we used wsj0-2mix, wsj0-3mix Hershey et al. (2016a),wsj0-4mix, ws0-5mix (Nachmani et al., 2020), Libri5Mix, Libri10Mix (Dovrat et al., 2021). The wsj0-2mix, wsj0-3mix, wsj0-4mix, and ws0-5mix datasets are made of 2, 3, 4, 5 speaker mixtures, respectively, created from the WSJ0 corpus. The datasets were created by exploiting randomly selected gains in order to achieve relative levels between 0 and 5 dB between the 2, 3, 4, 5 speech signals. The datasets are composed of 30 h training, 10 h validation, and 5 h test sets. The training and validation sets share common speakers, which is not the case for test set. Libri5Mix and Libri10Mix are speech mixture composed of 5 and 10 different speakers respectively. The dataset was created from the LibriMix dataset (Cosentino et al., 2020), which was created from LibriSpeech Panayotov et al. (2015). The mixtures were created in Dovrat et al. (2021)from clean utterances with no noise with the resulting mixtures having an SNRs that are normally distributed with a mean of 0 dB

and a standard deviation of 4.1 dB. The datasets wsj0-2mix, wsj0-3mix, wsj0-4mix, ws0-5mix, Libri5Mix and Libri10Mix assume that speech separation will happen in anechoic environment which is an unrealistic assumption for speech separation since mixed speech is always recorded by a distant microphone hence it is always riddled with noise and reverberation. To demonstrate the performance of the proposed technique to separate mixed speech which contain noise and reverberation, we use WHAM ! (Wichern et al., 2019) and WHARM ! (Maciejewski et al., 2020) datasets. These datasets were derived from wsj0-2mix dataset by adding environmental noise and noise+reverberation respectively. For all of these datasets, we used the test dataset for speech separation. The T-F bins or blocks generated from the speeches in the test dataset were processed by the relevant pre-trained model for embedding generation, e.g., if a pre-trained model was trained using time domain blocks it processes time-domain blocks to generate embeddings.

**Noise and Reverberation dataset**: To augment a given speech, we begin by adding noise to it then convolve the noisy speech with impulse responses (see figure 2 and 3). To add noise, we used the noise dataset from (Wichern et al., 2019). This dataset contains noise recorded with a binaural microphone in various areas in San Francisco Bay town. Each speech was mixed with a randomly sampled excerpt from this dataset. We ensured that the SNR between a given speech and noise in given mixture (noise+ speech) was less that 2. For reverberation we convolved the noisy speech with impulse responses developed in (Maciejewski et al., 2020). We used the medium impulse responses which simulate different acoustic conditions with reverberation times $T_{60}$ selected uniformly between 0.2 to 0.6 seconds.

**Downstream training:** During feature clustering, we used two configurations: First, we trained a Sep-Former on top of the frozen encoder to partition T-F bins or blocks. In the second configuration, the pre-trained encoder is fine-tuned during the clustering of spectrogram points or raw speech blocks. In this case, the pre-trained encoder is optimized in an end-to-end differentiable manner using equation 14 as the loss function. We therefore train four different SepFormer configurations: (i) SepFormer clustering of T-F bins based on frozen encoder trained using T-F bins (Sep1), (ii) SepFormer clustering of raw speech blocks based frozen encoder trained using raw speech blocks (Sep2), (iii) SepFormer clustering of T-F bins based on fine-tuning an encoder trained using T-F bins (Sep3) and (iv) SepFormer clustering of raw speech blocks based on fine-tuning an encoder trained using raw speech blocks (Sep4). We used the Adam optimiser and the cyclical learning rate (Smith, 2017) with a minimum learning rate of $1e-4$ and a maximum of $1e-1$. The models were trained with a single NVDIA V100 GPU for 2M steps with a batch size of 216 T-F bins or blocks. We optimized the downstream model according to equation 14 to generate partitions. We set the maximum number of clusters $k = 20$.

**Evaluation metrics**: We used objective metrics of Short-time objective intelligibility (STOI)(Taal et al., 2011), perceptual evaluation of speech quality (PESQ) algorithm (Rix et al., 2001), SI-SNR improvement (SI-SNRi), Signal-to-Distortion Ratio improvement (SDRi), Deep Noise Suppression MOS (DNSMOS) which is a reference-free metric that evaluates perceptual speech quality Reddy et al. (2021). It is a DNN based model trained on human ratings obtained by using an online framework for listening experiments based on ITU-T P.808. We also use SIG, BAK, OVRL: The non-intrusive speech quality assessment model DNSMOS P.835 (Reddy et al., 2022).

# 6 Results

## 6.1 Quality of clusters

To begin our experiments, we first evaluate which of the four model configurations (Sep1, Sep2, Sep3, Sep4) results in better cluster (partition) generation. We do this by measuring the quality of clusters generated by each. To evaluate how good the clusters are, we use the graph-based cluster measurement metrics proposed in (Yang & Leskovec, 2012). We are particularly interested in metrics that capture how well a given partition is separated from the rest i.e., quantifying the number of edges pointing from a given partition to other partitions. A good partition should have few edges pointing outwards. The most relevant metrics for our study being graph modularity and conductance.

**Cluster conductance (C)**$= \frac{c_s}{2m_s+c_s}$, if $S$ is a partition, the function C measures how similar the nodes of $S$ are where $m_s$ is the number of edges in $S$ i.e., $m_s = \{(u,v) \in E, u \in S, v \in S\}$ and $c_s$ is the number of edges in the boundary of $S$ i.e. $c_s = \{(u,v) \in E : u \in S, v \notin S\}$. Conductance quantifies the fraction of

edges pointing outside a given partition. Quality partitions should have a small conductance value.

**Graph modularity** $(\mathcal{Q}) = \frac{1}{4}(m_s - E(m_s))$ where $E(m_s)$ is the expected $m_s$. Quality partitions should have high modularity.

The results of our evaluation are reported in table 1. When raw speech block features are used to train the encoder, the pre-trained model generates more quality embeddings that lead to quality clusters in the downstream task as compared to when T-F bins are used as input. The features from pre-trained model with raw speech block as input generates better clusters in all the three test datasets used in evaluation. This is an indication that for this setup, the use of raw speech features captures more speech features as compared to the STFT transformed features. We also note that fine-tuning the encoder is beneficial to the downstream task as compared to the use of frozen encoder. Note that quality of cluster evaluation is done prior to speech reconstruction hence the drop in cluster quality when using STFT transformed features cannot be attributed to phase handling issues.

Table 1: Results of conductance C and modularity Q when using different input configurations and different mixtures. Here the values of C and Q have been multiplied by 100.

| WSJ0-3mix test-dataset | | |
|---|---|---|
| **Model Configuration** | **C($\downarrow$)** | $\mathcal{Q}(\uparrow)$ |
| Sep1 | 14.9 | 88.1 |
| Sep2 | 14.4 | 86.7 |
| Sep3 | 14.1 | 88.5 |
| Sep4 | 13.9 | 88.9 |
| **WSJ0-4mix test-dataset** | | |
| Sep1 | 16.2 | 85.3 |
| Sep2 | 15.8 | 86.9 |
| Sep3 | 15.1 | 87.2 |
| Sep4 | 14.6 | 87.6 |
| **WSJ0-5mix test-dataset** | | |
| Sep1 | 19.0 | 76.6 |
| Sep2 | 18.4 | 78.5 |
| Sep1 | 18.0 | 78.8 |
| Sep2 | 17.7 | 79.6 |

## 6.2 Performance on speech separation

We evaluated the quality of separated speeches by applying masks generated by Sep1, Sep2, Sep3 and Sep4 on wsj0-2mix, wsj0-3mix wsj0-4mix, ws0-5mix test datasets. Table 2 reports the results based on the evaluation metrics. In all the four datasets, the quality of separated speeches is highest on Sep4 based speech separation. This followed by that of Sep2. This trend does not follow the evaluation on cluster quality which indicated that Sep4 produced the best quality clusters followed by those of Sep3. We attribute the drop of quality in Sep3 to the phase reconstruction associated with STFT transformation. The results also show that when raw blocks are used in both pre-training and downstream task, the quality of separated speech is superior.

Table 2: Speech separation results when the two variants of inputs are used

| Model | SDRi($\uparrow$) | SI-SNRi($\uparrow$) | STOI($\uparrow$) | PESQ ($\uparrow$) | DNSMOS ($\uparrow$) | SIG ($\uparrow$) | BAK ($\uparrow$) | OVRL ($\uparrow$) |
|---|---|---|---|---|---|---|---|---|
| WSJ0-2mix test-dataset | | | | | | | | |
| Sep1 | 19.6 | 19.4 | 0.8969 | 3.93 | 3.98 | 3.96 | 4.11 | 4.01 |
| Sep2 | 21.9 | 21.6 | 0.9146 | 4.04 | 4.14 | 4.09 | 4.17 | 4.16 |
| Sep3 | 20.6 | 20.2 | 0.9054 | 3.98 | 4.02 | 4.00 | 4.13 | 4.07 |
| Sep4 | **22.6** | **22.3** | **0.9346** | **4.08** | **4.17** | **4.11** | **4.23** | **4.20** |
| WSJ0-3mix test-dataset | | | | | | | | |
| Sep1 | 17.9 | 17.3 | 0.8702 | 3.76 | 3.86 | 3.88 | 3.91 | 3.89 |
| Sep2 | 19.7 | 19.5 | 0.9051 | 4.01 | 4.05 | 4.01 | 4.08 | 4.11 |
| Sep3 | 18.6 | 17.9 | 0.8837 | 3.86 | 3.98 | 3.91 | 4.07 | 4.01 |
| Sep4 | **21.8** | **21.3** | **0.9150** | **4.04** | **4.10** | **4.07** | **4.14** | **4.09** |
| WSJ0-4mix test-dataset | | | | | | | | |
| Sep1 | 15.8 | 15.5 | 0.8414 | 3.52 | 3.67 | 3.75 | 3.81 | 3.78 |
| Sep2 | 17.4 | 17.0 | 0.8896 | 3.92 | 3.97 | 3.94 | 4.01 | 4.03 |
| Sep3 | 16.3 | 16.01 | 0.8522 | 3.70 | 3.84 | 3.87 | 4.03 | 3.98 |
| Sep4 | **20.9** | **20.5** | **0.9011** | **3.98** | **4.02** | **4.00** | **4.08** | **4.04** |

### 6.3 Comparison with other speech separation tools in few ( $n \leq 3$ ) source mixtures.

Here, we evaluate how the proposed technique's performance compares to other state of the art speech separation tools. The results are reported in table 3. In wsj0-2mix, Sep4 based speech separation improves SI-SNRi and SDRi of SepFormer+DM by 0.3 and SDRi by 0.5 respectively. In this dataset, sep4 achieves a SI-SNRi score of 22.6 which has a deviation of 1.5 from the best performing tool MossFormer2. With regards to SDRi, sep4 attains a score of 22.9 which has a deviation of 0.6 from the best performing tool TF-GridNet. In wsj0-3mix, the scalability of the proposed technique to high source mixtures is evidence as compared to the other tools. While the performance of SepFormer + DM with regard to SI-SNRi and SDRi drops by 2.8 and 2.7 respectively in the wsj0-3mix dataset when compared to its performance in wsj0-2mix, the performance of Sep4 based speech separation drops marginally by only 0.8 and 1.0 respectively in the two metrics. This is a signal of the ability of deep modularity technique to generalize to mixture with high sources. MossFormer2 SI-SNRi score drops to 22.2 in wsj0-3mix as compared to 24.1 in the wsj0-3mix dataset (i.e. a 1.9 drop). In wsj0-3mix dataset sep4 registers the second best SI-SNRi score of 21.8 which has a deviation of 0.4 from the best performing tool MossFormer2. The results show that the proposed unsupervised technique is competitive as compared to the supervised based state of the art speech separation techniques.

Table 3: Comparing the results of the proposed technique with other state of the art speech separation tools.

| Model | SI-SNRi | SDRi |
|---|---|---|
| **WSJ0-2mix test-dataset** | | |
| TF-GridNet Wang et al. (2023) | 23.5 | **23.6** |
| MossFormer2 Zhao et al. (2024) | **24.1** | - |
| SepFormer Subakan et al. (2021b) | 20.4 | 20.5 |
| SepFormer+DM Subakan et al. (2021b) | 22.3 | 22.4 |
| Wavesplit Zeghidour & Grangier (2021b) | 21.0 | 21.2 |
| Wavesplit+DM Zeghidour & Grangier (2021b) | 22.2 | 22.3 |
| DeepCASA Liu & Wang (2019b) | 17.7 | 18.0 |
| ConvTasnet Luo & Mesgarani (2019b) | 15.3 | 15.6 |
| Sep1 | 19.6 | 19.8 |
| Sep2 | 21.9 | 22.1 |
| Sep3 | 20.6 | 21.0 |
| Sep4 | 22.6 | 22.9 |
| **WSJ0-3mix test-dataset** | | |
| MossFormer2 | **22.2** | - |
| SepFormer | 17.6 | 17.9 |
| SepFormer+DM | 19.5 | 19.7 |
| Wavesplit | 17.3 | 17.6 |
| Wavesplit+DM | 17.8 | 18.1 |
| ConvTasnet | 12.7 | 13.1 |
| Sep1 | 17.9 | 18.1 |
| Sep2 | 19.7 | 19.7 |
| Sep3 | 18.6 | 19.0 |
| Sep4 | **21.8** | **21.9** |

### 6.4 Comparison with other speech separation tools in high( $n \geq 5$ ) source mixtures.

Here, we evaluate the performance of the proposed technique in mixtures with many sources. The results are shown in table 4. Sep4 based speech separation outperforms the existing tools by 0.2, 0.3 and 2.0 when evaluated on wsj0-5mix, Libri5Mix and Libri10Mix dataset on SDRi metric. This shows the proposed technique can scale to high source mixtures and generate quality estimated sources.

### 6.5 Results on WHAM! and WHAMR! datasets

We also evaluated the proposed technique in dataset that contains noise and reverberation. The results of the performance of the proposed technique on WHAM and WHARM datasets are shown in table 5. For WHAM dataset, the model performs denoising in addition to speech separation. In that case it must add a partition for noise. For WHARM dataset, the model performs denoising, dereverberation and speech separation. The results on the WHAM! and WHAMR! datasets are compared to the other state of the art tools. Speech

Table 4: Performance of the proposed technique on high source mixtures as compared to other tools that can perform high source mixtures separation.

| WSJ0-5mix test-dataset | |
|---|---|
| **Model** | **SDRi** |
| ConvTasNet Luo & Mesgarani (2019a) | 6.8 |
| DPRNN Luo et al. (2020) | 8.6 |
| MulCat Nachmani et al. (2020) | 10.6 |
| Hungarian Dovrat et al. (2021) | 13.2 |
| Sep1 | 10.4 |
| Sep2 | 12.3 |
| Sep3 | 10.9 |
| Sep4 | **13.4** |
| **Libri5Mix test-dataset** | |
| SinkPIT Tachibana (2021) | 9.4 |
| MulCat Nachmani et al. (2020) | 10.8 |
| Hungarian Dovrat et al. (2021) | 12.7 |
| Sep1 | 9.6 |
| Sep2 | 11.5 |
| Sep3 | 10.1 |
| Sep4 | **13.0** |
| **Libri10Mix test-dataset** | |
| SinkPIT Tachibana (2021) | 6.8 |
| MulCat Nachmani et al. (2020) | 4.8 |
| Hungarian Dovrat et al. (2021) | 7.8 |
| Sep1 | 7.9 |
| Sep2 | 8.1 |
| Sep3 | 9.3 |
| Sep4 | **9.8** |

separation based on Sep4 registers the best results in both datasets as compared to other state of the art results. This shows that the proposed technique is able to deal with mixtures with noise and reverberation.

Table 5: Comparing the results of the proposed technique with other state of the art speech separation tools on WHAM and WHARM test-dataset .

| WHAM test-dataset | | |
|---|---|---|
| **Model** | **SI-SNRi** | **SDRi** |
| SepFormer Subakan et al. (2021b) | 14.7 | 15.1 |
| SepFormer+DM Subakan et al. (2021b) | 16.4 | 16.7 |
| Wavesplit+DM Zeghidour & Grangier (2021b) | 16.0 | 16.5 |
| ConvTasnet Luo & Mesgarani (2019b) | 12.7 | - |
| Sep1 | 15.6 | 15.9 |
| Sep2 | 16.4 | 16.8 |
| Sep3 | 15.8 | 16.0 |
| Sep3 | 17.2 | 17.4 |
| **WHARM test-dataset** | | |
| SepFormer | 11.4 | 10.4 |
| SepFormer+DM | 14.0 | 13.0 |
| Wavesplit+DM | 13.2 | 12.2 |
| ConvTasnet | 8.3 | - |
| BiLSTM Tasnet | 9.2 | - |
| Sep1 | 13.4 | 14.0 |
| Sep2 | 14.1 | 14.4 |
| Sep3 | 13.6 | 14.3 |
| Sep3 | 15.0 | 15.4 |

## 6.6 Selecting similarity threshold $\theta$

Selecting the ideal threshold ($\theta$) when creating adjacency matrix is not trivial. If $\theta$ is high, we risk losing important relationships between frames. On the other hand, selecting low $\theta$ results in a large graph dominated by uninformative edges and increases the clustering time. To select the optimum $\theta$ we conducted experiments

with different datasets where we varied the value of $\theta$ and recorded modularity and the number of clusters generated. The graph showing how modularity and number of clusters generated vary when using ws0-5mix test dataset is shown in figure 5. The modularity values in figure 5 have been normalised by multiplying by 100, and values of number of clusters have been normalised by multiplying by 10 for easy visualisation. As can be seen in figure 5, as the similarity increases, modularity increases at the risk of generating a singleton partition. Decreasing the similarity lowers modularity and the risk of generating extra partitions increases. In our case we selected a $\theta = 0.3$. The same threshold is used in all the experiments.

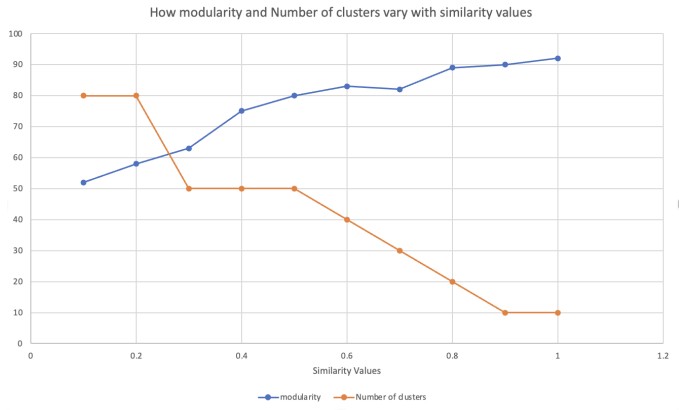

Figure 5: Graph showing how modularity and number of clusters vary as we change the similarity threshold

## 7 Ablation

### 7.1 $k-$means vs deep modularization

Here, instead of using deep modularization to generate partitions, we replace it with the classical $k$-means where we set $k = 20$. Concretely, after generating T-F bins or raw speech blocks features using a frozen pre-trained encoder, we exploit $k$-means to generate partitions which are then used for mask generation. We exploit $k$-means for the two versions:

1. $k$-means (raw speech blocks): Here $k$-mean is applied to cluster features of raw speech blocks.

2. $k$-means (T-F bins): Here $k$-mean is applied to cluster features of T-F bins.

The mask generation and speech reconstruction remains the same as described in section 4. We compare the results of quality of speech separated based on $k$-means and that of deep modularization. Since in $k$-means we use frozen pre-trained encoder to generate features of the input, we compare its results with that of Sep1 and Sep2 which also employ frozen encoder. The results are shown in table 6. The results show that deep modularization achieves superior results as compared to that of $k$-means. This shows that deep modularization is better at capturing relationship between T-F bins or raw speech blocks as compared to $k$-means.

### 7.2 Effect of input contamination during pre-training

Here, we investigate if training a pre-trained model with contaminated inputs is beneficial to the downstream task of speech separation. Therefore, we seek to compare separation results generated when the pre-trained encoder is trained on contaminated input vs clean input. To train an encoder on uncontaminated inputs, given a raw input of a clean speech signal $x \in R^T$, we generate spectrogram points (T-F bins) as described in section 3.1.1. We then design a function $f : O \mapsto R^d$ that maps the T-F bins to $d$-dimensional vectors by encouraging the representations of pairs of T-F bins from a given speaker to be closer to each other than the

Table 6: Comparing the results of the proposed technique with other state of the art speech separation tools.

| WSJ0-2mix test-dataset | | |
|---|---|---|
| Model | SI-SNRi | SDRi |
| Sep1 | 19.6 | 19.8 |
| Sep2 | 21.9 | 22.1 |
| $k$-means(raw speech blocks) | 17.5 | 17.8 |
| $k$-means(T-F bin) | 17.3 | 17.4 |
| **WSJ0-3mix test-dataset** | | |
| Sep1 | 17.9 | 18.1 |
| Sep2 | 19.7 | 19.7 |
| $k$-means(raw speech blocks) | 15.5 | 15.9 |
| $k$-means(T-F bin) | 15.1 | 15.7 |

representations of T-F bins from another random speaker( i.e, We do not apply any data augmentations to the T-F bins). Therefore, given $n$ clean speeches from $n$ different speakers, we transform the $n$ speeches into $n$ STFT representations and extract equally sized T-F bins from them. Let $\bar{X}$ denote the set of all T-F bins generated from the speeches $n$. Let the function $\mathcal{P}(.,. \mid \bar{X})$ be viewed as an augmentation pair generator such that it picks two pairs of T-F bins from $\bar{X}$ belonging to the same speaker. Given a batch of size $b$, for a positive pair $(x_i, x_i^+)$, we consider all the other $b - 2$ to be the negative examples. We train the model $f$ as described in section 3.1.1 with the loss defined in equation 5. For an input in time domain, given a time-domain speech signal $x \in R^T$, the signal is processed similar to the discussion in 3.1.2 to to generate a set of blocks $L \in R^{F \times W \times N}$. Here, $N$ represents the number of blocks generated. Given $n$ clean speech signals, the resulting set $\bar{X}$ of $N \times n$ blocks is used to train a model $f$ as described in section 3.1.2 with the loss defined in equation 5. We trained four different SepFormer configurations: (i) SepFormer clustering of T-F bins based on frozen encoder trained using uncontaminated T-F bins (USep1), (ii) SepFormer clustering of raw speech blocks based frozen encoder trained using uncontaminated raw speech blocks (USep2), (iii) SepFormer clustering of T-F bins based on fine-tuning an encoder trained using uncontaminated T-F bins (USep3), (iv) SepFormer clustering of raw speech blocks based on fine-tuning an encoder trained using uncontaminated raw speech blocks (USep4). The results of the proposed technique when the pre-trained model trained on non-augmented input vs that trained on augmented input when evaluated on the wsj0-2mix and the wsj0-3mix datasets is shown in table 7. For all the configurations, if the pre-trained model is trained on uncontaminated inputs the quality of separation drops on both SI-SNRi and SDRi. The same trend was noticed in all the datasets. This shows how important data contamination is important when training a pre-trained model that tailored for speech separation.

Table 7: Comparing the results of the proposed technique when pre-trained model is trained using augmented vs non-augmented inputs

| WSJ0-2mix test-dataset | | |
|---|---|---|
| Model | SI-SNRi | SDRi |
| Sep1 | 19.6 | 19.8 |
| Sep2 | 21.9 | 22.1 |
| Sep3 | 20.6 | 21.0 |
| Sep4 | **22.6** | **22.9** |
| USep1 | 19.2 | 19.4 |
| USep2 | 21.5 | 21.9 |
| USep3 | 20.1 | 20.8 |
| USep4 | 22.2 | 22.4 |
| **WSJ0-3mix test-dataset** | | |
| Sep1 | 17.9 | 18.1 |
| Sep2 | 19.7 | 19.7 |
| Sep3 | 18.6 | 19.0 |
| Sep4 | **21.8** | **21.9** |
| USep1 | 17.5 | 17.8 |
| USep2 | 19.2 | 19.5 |
| USep3 | 18.1 | 18.8 |
| USep4 | 21.3 | 21.5 |

## 8 Conclusion

This work proposes an unsupervised technique of speech separation. The technique relies on a pre-trained model to generate input features which are then used downstream. In the downstream task, we combine deep neural network and graph clustering objectives to create clusters of spectrogram points or raw speech blocks features. The clusters are created such that T-F bins or raw speech blocks dominated by a given speaker are clustered together. We conduct extensive experiments with a number of dataset and establish that the proposed tool achieves state of the art results.

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

## Impact Statement

"This paper presents work whose goal is to advance the field of Machine Learning. There are many potential societal consequences of our work, none which we feel must be specifically highlighted here.

# A   Appendix

