# OpenReview forum: "Speech Separation based on pre-trained model  and Deep Modularization"
_TMLR — Rejected by TMLR_

### Review · Reviewer_9mJu · 2024-04-17

**Summary Of Contributions:**

This paper presents an unsupervised training strategy for monaural speaker separation using a two-stage process. Initially, a deep neural network (DNN) trained with a contrastive loss extracts embeddings for each time-frequency (TF) unit (or blocks for the time-domain). Subsequently, these embeddings undergo clustering through an approach termed "deep modulation," which maximizes modularity (Newmen’06) by optimizing a partition assignment matrix using the SepFormer (Subakan et al.’21b) model coupled with a graph clustering objective (Muller’23).

**Audience:**

Yes

**Broader Impact Concerns:**

There is no concern on the ethical implications of the work.

**Claims And Evidence:**

Yes

**Requested Changes:**

Critical Changes:

1.	Revision of Mathematical Notations: The paper must thoroughly revise and define all mathematical symbols and notations used, ensuring consistency throughout the document.
2.	Update Benchmark Results: Update the benchmark results section to reflect current state-of-the-art models, such as TF-GridNet achieving 23.5 dB SI-SDR on WSJ0-2mix and MossFormer2 achieving 22.2 dB on WSJ0-3mix.

Recommended Enhancements:

1.	Expansion of Related Work Section: Broaden the discussion on related works to include other unsupervised methods like MixIt and WaveSplit, focusing on their training strategies and their approaches to resolving permutation ambiguity.

2.	Clarification of Dataset and Training Details: Provide explicit details on the datasets and noise sources used for training the pre-trained model, specifying whether real room impulse responses or simulations were used, and clarify if different models were trained for different datasets.
3.	Acknowledgment of Data Augmentation: Discuss the use of data augmentation in speech separation and how it compares to your method, especially highlighting dynamic mixing's role in enhancing performance.

**Strengths And Weaknesses:**

Strengths:
-The proposed unsupervised approach significantly improves performance over supervised methods that are trained with permutation invariant training (PIT) in scenarios involving more than three concurrent speakers. This shows that the proposed approach can effectively resolve the permutation ambiguity problem inherent in speech separation.

Weaknesses:

The paper suffers from inconsistent mathematical notation and undefined symbols, making it difficult to follow and understand. Here are few examples:

1) In Section 1, Symbol N is not defined (total number of time-frequency bins).
2) Page 4, Paragraph 1, symbol S is not defined (which I believe is the noisy and reverberant sample). On page 6, S is defined as a partition assignment matrix.
3) Page 5, Paragraph 1, a block L has the dimensions of FxSxN. S and N previously used for the noisy and reverberant sample and the total number of time-frequency bins, respectively.
4) In equation 6, symbol g is not defined.

In Section 1, the authors state that “PIT is also unable to handle the output dimension mismatch problem where there is a mismatch in the number of speakers between training and inference.” This assertion is not accurate. In supervised, talker-independent speech separation, it is possible to manage the output dimension mismatch. One can set the number of output layers to a large number and produce zero signal for inference if the number of speakers is smaller (See LibriCSS dataset).

In Section 2, the authors provide detailed comparisons between time-domain and frequency-domain approaches. They primarily list works that are supervised and utilize different input features. It would be beneficial for readers if the authors focused on training strategies or resolving permutation ambiguity in the related works. For example, the authors should mention other unsupervised methods for speech separation such as MixIt (Wisdom et al.’20) or WaveSplit (Zeghidour et al. '21) that estimate speaker embeddings to resolve permutation ambiguity. The authors also mention that time-domain speech separation approach perform better than DFT-based models as “these models address the key limitation of DFT-based models, since the models are designed to fully learn the magnitude and phase information of the input signal during training”. It is not correct. DFT-based models such as TF-GridNet (Wang et al.’23) can estimate both magnitude and phase via complex STFT and can achieve state-of-the-art results.

For training the pretrained model with contrastive loss, samples are taken from two sets: noisy and noisy+reverberant. In a batch of size b, a positive pair of noisy and noisy+reverberant would have b-2 negative examples. It is possible that negative and positive examples may share the same speaker. If so, what is the justification that the pretrained model can be trained effectively?

In Section 5, it is unclear what dataset is used for sampling noise and how the reverberation effect is generated, i.e., whether it is simulated or based on real room impulse responses. Did you use the WHAM! training data? If so, please explicitly mention it. Also, it is not clear whether different SepFormers were trained for each individual dataset.

The state-of-the-art speech separation mentioned in the paper is outdated. For WSJ0 2-mix and 3-mix, several models have achieved much better results than SepFormer+DM. TF-gridNet achieved 23.5 dB SI-SDR for WSJ-2mix, while MossFormer2 reached 22.2 dB on WSJ-3mix.

Most models benchmarked on WSJ2-3 mix use only the official training dataset without any data augmentation. Data augmentation or dynamic mixing (DM) has been shown to significantly improve performance. Please acknowledge this and draw a comparison with your own setup.

---

### Review · Reviewer_dsEU · 2024-04-28

**Summary Of Contributions:**

This paper proposes a training framework for DNN-based speech separation based on unsupervised clustering. In the literature, speech separation with clustering, known as Deep Clustering [Hershey et al. (2016b)], has been developed with supervised training. This paper extends Deep Clustering toward unsupervised settings by exploiting a graph clustering technique called deep modularization. Experimental results on several benchmarks suggest that the proposed training framework works comparable to or better than supervised training.

**Audience:**

Yes

**Claims And Evidence:**

No

**Requested Changes:**

Major requirements:
- Motivation for unsupervised training should be clarified if the authors require clean speech for pre-training of the encoder.
- Performance of STFT-domain nonnegative masking methods, e.g., [Wang et al. (2018)] or [a], and oracle masks, e.g., ideal binary masks or phase-sensitive approximation, should be added to WSJ0-2mix results. Comparison of Sep1,3 and the aforementioned methods helps to clarify the benefit of the proposed training framework on the same separation scheme, i.e., STFT-domain nonnegative masking.
- MixIT should also be added for experimental evaluation to clarify the advantages of the proposed method when comparing existing unsupervised training frameworks.

Minor requirements:
- Figs. 2 and 3 seem to indicate that reverberation is convolved with a monaural noisy signal. This process is inappropriate for a model of actual noisy reverberant signals. In detail, applying the same reverberation to the speech and noise means that both speech and noise sources are in the same position, which is not realistic. The author need to fix the pre-processing part or at least explain the motivation of the order of pre-processing.
- The font size in Fig. 5 should be larger for visibility.
- After some equations, e.g., Eq. (1), the following lines start with "Where". They should be fixed to "where".
- In P. 2, it needs a space between "Takahashi et al. (2019)" and "Neumann et al.".

**Strengths And Weaknesses:**

Strengths:
- The proposed method does not explicitly require the pair of the mixture and its ingredients.
- The proposed method performs well even in an out-of-domain condition, i.e., WSJ for pre-training and LibriMix for separation.

Weakness:
- The pre-training of the encoder requires a clean speech dataset, e.g., WSJ is used in the paper. If clean speech datasets are available, the motivation for unsupervised training is ambiguous.
- Performance for the proposed STFT domain non-negative masking (Sep1 and Sep3) is surprisingly good. For instance, [Wang et al. (2018)] showed that SDRs for the ideal binary mask and the ideal phase-sensitive approximation are 13.5 dB and 16.4 dB, respectively. While SDR and SDRi are different, and the authors applied MISI as post-processing, the performance for Sep3 (SDRi of 21.0 dB) seems to be too high as non-negative masking with MISI. The reason for such a performance is unclear. Note that, to the best of this reviewer's knowledge, SOTA for supervised STFT domain non-negative masking on WSJ0-2mix is SDRi of 15.6 dB [a].
- While the authors mentioned MixIT as an existing unsupervised training framework, MixIT was not compared with the proposed method in experimental evaluation.

[a] @INPROCEEDINGS{8683231,
  author={Wang, Zhong-Qiu and Tan, Ke and Wang, DeLiang},
  booktitle={ICASSP 2019 - 2019 IEEE International Conference on Acoustics, Speech and Signal Processing (ICASSP)},
  title={Deep Learning Based Phase Reconstruction for Speaker Separation: A Trigonometric Perspective},
  year={2019},
  volume={},
  number={},
  pages={71-75},
}

---

### Review · Reviewer_rAx9 · 2024-05-28

**Summary Of Contributions:**

The paper attempts to describe an unsupervised speech separation method, but fails. No significant contribution is presented.

**Audience:**

No

**Broader Impact Concerns:**

N/A.

**Claims And Evidence:**

No

**Requested Changes:**

I recommend a reject decision and I do not want to encourage resubmission, so I would not request any changes.

**Strengths And Weaknesses:**

Strengths:
I see no strengths.

Weaknesses:
1. The paper is badly written and hard to read due to lack of clarity in both text as well as in its mathematical presentation.
2. It is not clear if the networks introduced in the paper take as input a signal, a time-frequency patch or a single time-frequency bin. At times, it feels like it takes a signal, at other times a single time-frequency bin and the math does not make it clearer since an x or an f is used without making it clear what they mean.
3. Even assuming the best guesses for the unclear methods, the introduced method does not have much novelty and is not too far from "deep clustering" which was an earlier method later abandoned due to superior methods introduced later.
4. The claimed results in the paper do not make sense since the method finds a binary mask (as my best guess due to lack of clarity) and even an oracle binary mask would not achieve those performance metrics. And no, using the MISI algorithm will not improve a binary mask output.
5. The paper talks about a time-frequency bin as if it is equivalent to a block of encodings from the time-domain approach. A time-frequency bin is a single bin, but the encoding blocks seem larger, but it is not totally clear since we do not know what the frame lengths are.
6. In the augmentation, first adding noise and then reverberating the sum does not make sense since speech and noise do not go through the same reverberation typically.
7. The pretrained model pools its outputs and obtains a 1280 dimensional output from its input. It looks like a Sepformer takes this output (f) and provides k softmax outputs. Now, a Sepformer requires a sequence input since it is a dual-path transformer model that operates on sequences of data. If f is a 1280 dimensional vector which is processed by a sepformer, what is the sequence input here?
8. It looks like we would have to know the number of speakers in a mixture to run this method since the output is k values and k must equal the number of speakers. Assuming knowing the number of speakers is not possible in practice and this assumption makes things invalid.
9. The method is not "unsupervised" even though it claims to be. Being unsupervised means using audio clips where we do not know whether they contain a single speaker or not. Using single speaker clips makes a method supervised. We can easily mix those single speaker clips to make supervised examples. The method may not mix them, and use them in an "unpaired" way, but it does not make it unsupervised.
10. Comparing with k-means with k=20 clusters does not make sense. How can you obtain 2 or 3 masks from k=20 clusters? This is not clear and not explained in the paper.
11. How can one obtain a non-binary mask? It is said that the authors obtain k masks in the range [0,1] and then it is mentioned that the mask is either 1 or 0.
12. The authors mention they use a decoder (transposed encoder) to obtain waveforms in the time-domain method. What is meant by the transposed encoder? Usually these methods train a separate decoder and there is no guarantee that a transposed encoder would work to reconstruct the waveform.

---

### Decision · Action_Editor_up71 · 2024-06-24

**Recommendation:** Reject

**Comment:**

This paper received two initial reviews that were somewhat negative, but made constructive suggestions on how the paper could be improved, and one more strongly negative review. The authors replied to one reviewer, but not to the others, and do not appear to have submitted a revision of the paper. There really is no option here but to reject the paper.

**Audience:**

Speech separation is a topic of interest to at least part of TMLR's readership.

**Claims And Evidence:**

The consensus among the reviewers is that the claims of the paper are not sufficiently well supported by accurate, clear, and convincing evidence.
- Reviewers dsEU and rAx9 both point out that the paper is claiming SDRi results for non-negative masking that exceed oracle performance. The authors did not respond to this statement.
- Reviewers dsEU and rAx9 both argue that a speech separation method that requires clean, single-speaker clips for training is not unsupervised. The knowledge that a clip contains only one speaker is a strong form of supervision for speech separation. The authors did not respond to this statement.
- All reviewers raised concerns about the clarity of the paper in various aspects, including mathematical notation, the description of the model and how it operates, and the discussion of related work, but no revision addressing these concerns was provided.

**Resubmission Of Major Revision:**

The authors may consider submitting a major revision at a later time.